# Linear Dynamical Systems as a
# Core Computational Primitive

**Shiva Kaul**
Computer Science Department
Carnegie Mellon University
Pittsburgh, PA 15213
skkaul@cs.cmu.edu

## Abstract

Running nonlinear RNNs for $T$ steps takes $\Omega(T)$ time. Our construction, called LDStack, approximately runs them in $O(\log T)$ parallel time, and obtains arbitrarily low error via repetition. First, we show nonlinear RNNs can be approximated by a stack of multiple-input, multiple-output (MIMO) LDS. This replaces nonlinearity across time with nonlinearity along depth. Next, we show that MIMO LDS can be approximated by an average or a concatenation of single-input, multiple-output (SIMO) LDS. Finally, we present an algorithm for running (and differentiating) SIMO LDS in $O(\log T)$ parallel time. On long sequences, LDStack is much faster than traditional RNNs, yet it achieves similar accuracy in our experiments. Furthermore, LDStack is amenable to linear systems theory. Therefore, it improves not only speed, but also interpretability and mathematical tractability.

## 1 Introduction

Nonlinear RNNs have two crucial shortcomings. The first is computational: running an RNN for $T$ steps is a sequential operation which takes $\Omega(T)$ time. The second is analytical: it is challenging to gain intuition about the behavior of a nonlinear RNN, and even harder to prove this behavior is desirable. These shortcomings have motivated practitioners to abandon RNNs altogether and to model time series by other means. These include hierarchies of (dilated) convolutions [Oord et al., 2016, Gehring et al., 2017] and attention mechanisms which are differentiable analogues of key-value lookups [Bahdanau et al., 2014, Vaswani et al., 2017]. In these models, the underlying parallel primitives are convolution and matrix multiplication, respectively.

This paper addresses both of these shortcomings. We present a method to approximately run and differentiate nonlinear RNNs in $O(\log T)$ parallel time, by rebuilding them from linear dynamical systems (LDS). In these, the next state $s_{t+1} = As_t + Bx_t$ is a linear function of the current state $s_t$ and input $x_t$. They are a mainstay of control theory and many engineering applications because their behavior can be understood and regulated [Zhou et al., 1996]. Single-input, multiple-output (SIMO) LDS, which map a sequence of input numbers to a sequence of output vectors, are our core primitive: we present an algorithm to run and differentiate them in $O(\log T)$ parallel time.

**Summary of Main Ideas**. Our approach is to (1) approximate the RNN by a stack of multiple-input, multiple output (MIMO) LDS, then (2) approximate the MIMO LDS by an aggregation of single-input, multiple-output (SIMO) LDS, and finally (3) run the SIMO LDS in $O(\log T)$ parallel time using scans and reductions. In step (1), we take the LDS, measure the deviations of its linear steps from desired nonlinear ones, and add those as corrections to the LDS in the subsequent layer. This scheme is naturally parallel, since the corrections are based on only local information; surprisingly, it is provably consistent. A multiplicative variant has already been extensively used to analyze nonlinear, continuous-time dynamical systems [Tomás-Rodríguez and Banks, 2010].

For step (2), we consider two kinds of aggregation: averaging and concatenation. The averaging approach uses a standard technique in randomized numerical linear algebra: the $d$-dimensional inputs $x_t$ are repeatedly, randomly projected to a single dimension. The concatenation approach pre-applies a $d \times d$ transformation to the inputs. Then, the inputs are given to $d$ coupled SIMO LDS, each of size $n/d$. This approach builds upon the canonical form of Luenberger [1967], which decomposes the MIMO LDS into smaller SIMO LDS, whose sizes are called the *controllability indices* of the MIMO system. Unfortunately, these quantities are onerous to estimate or to even compute. Using a perturbed Luenberger form, we show that a uniform size $n/d$ may be used with essentially no loss in generality.

Finally, step (3) exploits the linear-algebraic structure of SIMO LDS. It is known that linear recurrences $s'_{t+1} = \lambda \circ s'_t + b_t$, which involve entrywise multiplication $\circ$, can be run in $O(n \log T)$ parallel time via scans and reductions. A SIMO LDS can be taken to this form via diagonalization, i.e. by running the LDS in the basis of its eigenvectors. When the SIMO LDS is in a canonical form, its eigenvectors have closed-form expressions in terms of its eigenvalues. Accordingly, the set of SIMO LDS is exactly parameterized by just $n$ numbers, which are provided to the recurrence solver.

**Outline**. We present our approach in a bottom-up fashion. Then, we empirically evaluate it on artificial and real datasets. LDS achieve state-of-the-art performance on the copy memory problem. LDStack can be substantially faster than traditional RNNs, while achieving competitive accuracy. Finally, we offer guidance on how our constructions could be improved in future work.

## 2  Linear Dynamical Systems

Linear dynamical systems have enjoyed a renaissance in machine learning theory. There have been many recent advances in algorithms for learning LDS from input-output data [Hardt et al., 2016, Oymak and Ozay, 2019, Simchowitz et al., 2019, Sarkar and Rakhlin, 2019]. The sample complexity of this task is well-studied [Simchowitz et al., 2018, Jedra and Proutiere, 2019]. As analytical testbeds, they capture the behavior of optimization algorithms [Lessard et al., 2016] and establish baseline performance for reinforcement learning [Recht, Matni et al., 2019] and online learning [Hazan et al., 2017, Kozdoba et al., 2019, Ghai et al., 2020]. Efficient and robust algorithms have recently been developed for controlling LDS [Dean et al., 2019, Hazan et al., 2020].

This section reviews some basic material about LDS. At time $t \in [T]$, let the input be $x_t \in \mathbb{R}^d$. Starting from an initial state $s_0 \in \mathbb{R}^n$, an LDS produces subsequent states $s_{t+1}$:

$$s_{t+1} = As_t + Bx_t = A^{t+1}s_0 + \sum_{\tau=0}^{t-1} A^{\tau+1}Bx_{t-\tau} \qquad y_t = Cs_t + Dx_t + D_0 \qquad (1)$$

where $A \in \mathbb{R}^{n \times n}$ and $B \in \mathbb{R}^{n \times d}$. By recursively unrolling the first equality, we see the states are a convolution of the inputs (with an infinite kernel size and only one stride dimension). Outputs $y_t \in \mathbb{R}^m$ may be optionally produced, using $C \in \mathbb{R}^{m \times n}$, $D \in \mathbb{R}^{m \times d}$, and $D_0 \in \mathbb{R}^m$.

### 2.1  SIMO Canonical Form

An LDS is *reachable*, roughly speaking, if we can take it to any state by supplying the right input.

**Definition 1** (Reachability). *A state $s \in \mathbb{R}^n$ is reachable if there is a sequence of inputs $x_1, \ldots, x_T$ which leads to $s_T = s$. An LDS is reachable if every state $s \in \mathbb{R}^n$ is reachable.* [1]

**Lemma 1** (Hautus). *An LDS is reachable iff $A$ is nonsingular and, for all $\gamma \in \mathbb{C}$, the $n \times (n + d)$ matrix $[\gamma I - A; B]$ has full rank $n$.*

A reachable SIMO LDS $(\tilde{A}, \tilde{B}, \tilde{C}, D)$ is placed in canonical form $(A, B, C, D)$ by $\mathcal{T} \in \mathbb{R}^{n \times n}$:

$$A = \mathcal{T}\tilde{A}\mathcal{T}^{-1} = \begin{pmatrix} 0 & 0 & 0 & -a_0 \\ \ddots & 0 & 0 & \vdots \\ 0 & 1 & 0 & -a_{n-2} \\ 0 & 0 & 1 & -a_{n-1} \end{pmatrix} \qquad B = \mathcal{T}\tilde{B} = \begin{pmatrix} 1 \\ 0 \\ \vdots \\ 0 \end{pmatrix} \qquad C = \tilde{C}\mathcal{T}^{-1} \qquad (2)$$

$\mathcal{T}^{-1}$ is the controllability matrix of $(\tilde{A}, \tilde{B})$ [Ding, 2010], which will be defined in (4). $a_0, \ldots, a_{n-1}$ are the coefficients of $A$'s characteristic polynomial $t \mapsto t^n + \sum_{i=0}^{n-1} a_i t^i$. $A$ is determined by its eigenvalues $\lambda$, since $a_i = (-1)^{n-i} e_{n-i}(\lambda)$, where $e_i$ is the $i$th elementary symmetric polynomial. Equation (2) is called the Frobenius companion form, and is one of many similar companion forms [Fiedler, 2003, Eastman et al., 2014]. We also consider the transpose form, which replaces $(A, B)$ by $(A^T, [0, \ldots, 0, 1]^T)$. In these forms, the number of parameters reduces from $n^2 + n$ to just $n$, for $\lambda$.

## 2.2 Diagonalization

$A = V^{-1} \Lambda V$ where $\Lambda$ is a diagonal matrix of the eigenvalues $\lambda$. $V$ is the Vandermonde matrix in $\lambda$ with entries $V_{i,j} = \lambda_i^{j-1}$. Its rows are the (row) eigenvectors of $A$. Since $A$ is not symmetric, the eigenvectors are neither real nor orthonormal. However, since $A$ is real, any complex eigenvalues come in conjugate pairs: if $\lambda_j = \alpha_j - \beta_j i$ is an eigenvalue, then so too is $\overline{\lambda}_j = \alpha_j + \beta_j i$. Defining $s'_t = V s_t$, $B' = VB$ and $C' = CV^{-1}$, we diagonalize the system to a *modal* form:

$$s'_{t+1} = VAs_t + VBx_t = \lambda \circ s'_t + B'x_t \qquad y_t = C's'_t + Dx_t + D_0 \qquad (3)$$

The transpose form is often factored in a slightly different way, for analytical purposes.

**Lemma 2.** $A = U\Lambda U^{-1}$ where the $j$th column of $U$ is $u_j = \left[ \frac{1}{\lambda_j^{n-i}} \right]_{1 \le i \le n}$. *(Leslie [1945], Brand [1964]; see the appendix for a self-contained proof.)*

Multiplication by $V$ and $V^{-1}$ are equivalent to polynomial evaluation and interpolation, respectively. That is, $Vc$ evaluates a univariate polynomial, with coefficients $c$ in the monomial basis, at points $\lambda_1, \ldots, \lambda_n$; $V^{-1}y$ recovers the coefficients. Naively performing these operations may be numerically unstable, due to high-degree powers of $\lambda$. These operations may be more accurately performed in $O(n^2)$ time by Horner's method and the algorithm of Björck and Pereyra [1970], respectively.

## 2.3 MIMO Luenberger Form

Let $b_i$ be the $i$th column of $B$. The controllability matrix of a MIMO LDS has dimensions $n \times (n \cdot d)$:

$$\mathcal{C} = [b_1, \ldots, b_d, Ab_1, \ldots, Ab_d, \ldots, A^{n-1}b_1, \ldots, A^{n-1}b_d] \qquad (4)$$

From left to right, take $n$ columns, but skip a column if it is linearly dependent on the columns taken so far. If this procedure skips $A^u b_i$, it will also skip the higher powers $A^{u+1} b_i$. For $i \in [d]$, the *controllability index* $\mu_i$ is the first power of $A$ skipped for $b_i$. For reachable LDS, $\sum_i \mu_i = n$.

The Luenberger form $(A^{*d}, B^{*d}E, C, D)$ expresses any reachable, multiple-input LDS as the concatenation of $d$ coupled, reachable, single-input LDS, whose sizes equal the controllability indices [Luenberger, 1967]. Visual examples of $A^{*d}$ and $B^{*d}$ are given in Figure 3. $A^{*d}$ has, along the block diagonal, $d$ transpose-form SIMO LDS transition matrices of sizes $\mu_i$. It has off-diagonal entries which couple the SIMO LDS at their inputs. Similarly, $B^{*d}$ is the block diagonal matrix of $d$ transpose-form $B$ vectors, each of dimension $\mu_i \times 1$. $E$ is an invertible, upper triangular matrix which depends on the original system parameters. It is pre-applied to the inputs.

# 3 SIMO LDS in $O(n \log T)$ Parallel Time and $n$ Parameters

The following result makes reachable SIMO LDS our key computational primitive.

**Proposition 1.** *Reachable, SIMO, $n$-state LDS are exactly represented by their distinct, nonzero, complex eigenvalues $\lambda \in \mathbb{C}^n$, without further constraints. These eigenvalues can be concretely parameterized by $n$ (or fewer) real numbers. Given the parameters and a length-$T$ sequence of inputs $x$, it is possible to compute the LDS outputs, and their gradients with respect to the parameters, in $O(n \log T + n^2)$ time on $O(T)$ parallel processors.*

It is underpinned by the following algorithm for parallel linear recurrences (PLR).

**Proposition 2.** *Let $\lambda_1, \ldots, \lambda_T$ and $b_1, \ldots, b_T$ be sequences of $n$-dimensional vectors. Let $\circ$ denote entrywise product between vectors. For $t \in [T]$, the recurrence $s'_{t+1} = \lambda_t \circ s'_t + b_t$, and its gradients, can be computed in $O\left(n\left(\frac{T}{p} + \log p\right)\right)$ depth (aka parallel time) on $p$ parallel processors. This is $O(n \log T)$ parallel time when $p = O(T)$. [Martin and Cundy, 2018]*

1. Initialize real variables and use them to define eigenvalues $\lambda$. In the standard parameterization (left), the variables are $\alpha$ and $\beta$, whose total length is $n$. In the unit parameterization (right), the variables are $\theta$, whose length is $n/2$.

$$a \sim \text{Normal}(0, 1/n)^n \qquad\qquad \theta \sim \text{Uniform}(-2\pi, 2\pi)^{n/2}$$

$$\lambda = \text{roots}\left(t \mapsto t^n + \sum_{i=0}^{n-1} a_i t^i\right) \qquad \lambda = [\exp(\theta i), \exp(-\theta i)]$$

$$\alpha, \beta \text{ satisfy } \lambda = [\alpha + \beta i, \alpha - \beta i]$$

2. Given a sequence of inputs $x \in \mathbb{R}^T$, compute the sequence of states $s'_{t+1}$, and their gradients $\nabla s'_{t+1}$ with respect to the underlying real parameters. Use the algorithm of Proposition 2 on the recurrence $s'_{t+1} = \lambda \circ s'_t + B' x_t$ given in (3), where $B'$ is the all-ones vector.

3. (Optional). Convert $s_t = V^{-1} s'_t$ using the algorithm of Björck and Pereyra [1970]. Finally, compute the outputs $y_t$ using an additional dense layer, as in Equation (1). Alternatively, compute $y_t = \text{Re}(C' s'_t) + D x_t + D_0$ using a relaxation $C' \in \mathbb{C}^{m \times n}$.

Figure 1: Summary of how reachable SIMO LDS, with spectral parameterizations, can be used as a fast layer in a neural network. Also consider the "hinge" parameterization in the appendix. Martin and Cundy [2018] implemented the PLR algorithm in CUDA; we extend it for complex inputs.

Proposition 1 involves three steps. First, the complex LDS eigenvalues $\lambda$ must be concretely parameterized by real numbers, which in turn must be reasonably initialized. Then, the LDS must be diagonalized according to (3). At first glance, it seems more straightforward to directly parameterize $\lambda$ and $B'$ in the diagonal form (3). Unfortunately, this does not exactly capture the set of reachable SIMO LDS, unless additional constraints are imposed. If $\lambda$ and $B'$ are taken to be real, then only a subset is expressed; if they are complex, then a superset is expressed, and the number of parameters doubles. For analytical and practical reasons, it is desirable to exactly use reachable LDS. (For example, if LDS are stacked in a neural network, then reachability would ensure each layer can supply a full spectrum of input to the subsequent layer.)

**Parameterization**. The standard approach is to separately parameterize the real and imaginary (if present) parts of $\lambda$. Since the complex eigenvalues present in conjugate pairs, this requires only $n$ real parameters $(\alpha, \beta)$ in total. More specifically, the complex pairs are $\lambda_j = \alpha_j - \beta_j i$ and $\bar{\lambda}_j = \alpha_j + \beta_j i$. The real eigenvalues just have $\alpha_j$. For long-term dependencies, it is useful to constrain $|\lambda_j| = 1$, as in orthogonal or unitary $A$ [Arjovsky et al., 2016]. This constraint is trivial in our framework. Suppose $\lambda_j$ has polar representation $(r_j, \theta_j)$. Then a zero real part of $\ln \lambda_j = \ln r_j + \theta_j i$ corresponds to magnitude $r_j = 1$. Parameterize $\ln \lambda$ with 0 real part and $\pm \theta$ imaginary part, then exponentiate.

**Initialization**. For the previously defined real variables, typical random initialization, such as sampling from a truncated normal, lead to numerical instability. In the standard parameterization, we found it useful to initialize near unit eigenvalues. It is known that a monic polynomial with random coefficients has roots $\lambda$ of magnitude close to 1 [Hughes and Nikeghbali, 2008]. These may be obtained by randomly initializing the coefficients $a$ in (2), and then computing the eigenvalues of $A$ [Aurentz et al., 2015]. For the unit parameterization, the coordinates $\theta_j$ must be kept numerically distinct. For moderate $n$, uniform random initialization is suitable. For large $n$, a low-discrepancy sequence, such as the van der Corput sequence, may be preferable.

**Diagonalization**. The two computational tasks are computing $B'$ (for use in PLR) and converting between $s_t$ and $s'_t$. For the standard form, $B' = V[1, 0, \ldots, 0]^T = [1, \ldots, 1]$ since that is the first column of $V$. As reviewed in Section 2, conversion between $s_t$ and $s'_t$ may be accomplished by polynomial evaluation and interpolation algorithms. For the transpose form expressed in terms of $U$, $B'$ is the last column of $U^{-1}$. For completeness, this is derived in the appendix.

**Lemma 3.** *Given the (unnormalized) definition of $U$ in Lemma 2, the complex conjugate of the last column of $U^{-1}$ is $B' = \left[\lambda_i^{n-1} / \prod_{j \neq i} (\lambda_i - \lambda_j)\right]_{1 \leq i \leq n}$*

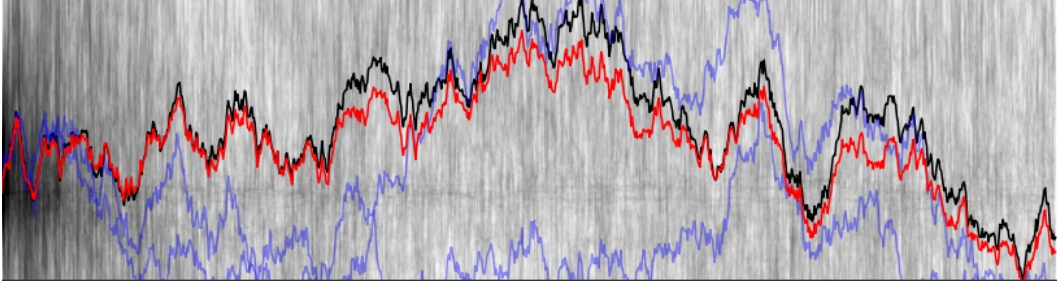

Figure 2: Illustration of a MISO LDS (black), of state size $n = 16$, operating on inputs of $d = 32$ dimensions over $T = 1024$ timesteps, approximated by SISO LDS. In light gray (nearly filling the background) are 512 SISO LDS, induced by random projections per Proposition 6. These have very high variance and do not approximate the MISO LDS. The two blue lines represent the average of two independent subsamples of 16 SISO LDS. These small averages still do not approximate the MISO LDS. The red line is the average of all 512 SISO LDS. This is fairly close to the MISO LDS.

**Related Work**. LDS are often reparameterized for computational benefit [Shalit and Chechik, 2014], sometimes in terms of induced subspaces De Cock and De Moor [2002], Huang et al. [2017]. Chang et al. [2018] also study complex eigenvalue parameterizations with zero real part. Hsu et al. [2020] analyze LDS clustering using the Vandermonde decomposition. Previous algorithms attempt to run LDS in constant time with respect to $T$ [Martens, 2010, Kozdoba et al., 2019]. However, these works rely on stability assumptions and approximations: they do not exactly compute forward and backward passes of LDS. Furthermore, they require the inputs to be partially and completely noise, respectively. Surprisingly, Lemma 3 does not plainly appear in the literature, even in recent work on generalizations of Vandermonde matrices [Rawashdeh, 2018]. Its proof uses the same technique as the "eigenvectors from eigenvalues" theorems that have gained recent attention in disparate areas of applied mathematics [Denton et al., 2019]. These results are more general, but do not yield closed-form expressions, and do not directly apply to the inverse matrix $U^{-1}$.

## 4 Approximating MIMO LDS by SIMO LDS

### 4.1 Improper Learning: Random Projection

MIMO LDS can be approximated by the average of $r$ SIMO LDS, each produced by randomly projecting the input vectors to a single dimension. These LDS share the same weights $\lambda$.

**Proposition 3.** *Let $x_1, \ldots, x_T \in \mathbb{R}^d$ and $y_1, \ldots, y_T \in \mathbb{R}^m$ be the inputs and outputs of a reachable MIMO LDS with parameters $(A, B, C, D)$. For each $j \in [r]$, let $g_j$ be a $d$-dimensional standard normal vector, $x_t^{[j]} = x_t^T g_j$ be projected scalar inputs, and $(A, Bg_j, C, D)$ be the parameters of a SIMO LDS producing outputs $y_t^{[j]}$. Let $\hat{y}_t = \frac{1}{r} \sum_{j=1}^{r} y_t^{[j]}$ be the average output. For each $t \leq T$, $\mathbf{E} \, ||y_t - \hat{y}_t||^2 = \sum_{j=1}^{m} 2 \, ||Z_{t,j}||_F^2 \, /r$, where $Z_{t,j} = \sum_{\tau=1}^{t-1} x_{t-\tau} C_{j,:} A^\tau B$. Furthermore, the SIMO LDS are almost surely reachable, and share the same canonical form matrix.*

The proof of this equality uses standard techniques. Here is some brief intuition for the result. Suppose $m = 1$ and each $x_t$ has standard $N(0, 1)$ components, as is typical in dynamical systems literature. Also assume that $A$'s spectral radius $\rho < 1$ (i.e. the LDS is *strictly stable*), $||B||_2 \leq 1$, and $||C|| \leq 1$. By the definition of the Frobenius norm and independence of each input:

$$\mathbf{E} \, \text{tr}(Z_t^T Z_t) = \text{tr} \sum_{\tau=1}^{t-1} B^T A^{\tau T} C^T \left( \mathbf{E} \, x_{t-\tau}^T x_{t-\tau} \right) C A^\tau B \leq d \sum_{\tau=1}^{t-1} \rho^{2\tau} \leq d \frac{\rho^2}{1 - \rho^2} \quad (5)$$

**Related Work**. Gaussian projections are a key technique in randomized algorithms [Johnson and Lindenstrauss, 1984, Kannan and Vempala, 2017]. Model reduction is the approximation of large-size LDS by smaller-size LDS [Antoulas, 2005]. Proposition 3 does not reduce the size of the LDS, but rather the dimension of its inputs.

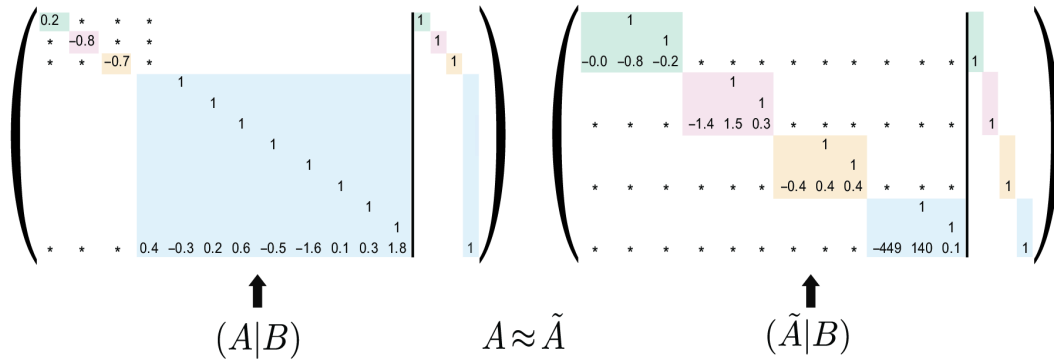

Figure 3: Left: is the Luenberger canonical form of a multiple-input LDS, with $A$ to the left of the vertical line and $B$ to the right. It decomposes into four single-input LDS of sizes $9, 1, 1$ and $1$, which match the controllability indices. After the addition of a tiny amount of noise (in the form of a Gaussian matrix with variance $0.00000001$), the canonical form decomposes into evenly-sized single-input LDS. The asterisks denote nonzero values which couple the single-input LDS.

## 4.2 Proper Learning: Perturbed Luenberger Form

Proper learning of LDS, also known as system identification, is the task of recovering the parameters $(A, B, C, D)$ from input-output data. The Luenberger form, reviewed in Section 2.3, exactly decomposes a MIMO LDS into a concatenation of smaller, SIMO LDS. It establishes a promising connection between proper learning of MIMO LDS and proper learning of SIMO LDS. However, as a parameterization used during learning, it has a crucial problem: the controllability indices, defining the sizes of the SIMO LDS, are not known. In practice, the SIMO LDS must be sized according to a loose upper bound, which then makes learning improper. Fortunately, the following result shows that any MIMO LDS is nearly equal to a concatenation of coupled SIMO LDS, each of known size.

**Proposition 4.** *Let $n$ be divisible by $d$. Let $(A, B)$ be the parameters of a reachable size-$n$ LDS taking $d$-dimensional inputs. For any $\epsilon > 0$, there exists a perturbed system $(\tilde{A}, B)$ such that (1) $||A - \tilde{A}|| \leq \epsilon$, and (2) the controllability indices of $(\tilde{A}, B)$ are all $n/d$. Therefore, the Luenberger form of $(\tilde{A}, B)$ is a concatenation of $d$ coupled SIMO LDS, each of size $n/d$.*

We may effectively treat any MIMO LDS data as if it originated from a system with equal controllability indices, i.e. equally-sized SIMO LDS. This result suggests that proper learning of LDS is largely equivalent to proper learning of SIMO LDS, which supports the latter's consideration as a key primitive. We present the perturbed Luenberger form as a conceptual reduction from MIMO to SIMO, rather than a practical algorithmic tool. The practical issue is that the SIMO LDS are coupled: the next state for each LDS depends on not just its own state, but also on the state of the other $(n/d) - 1$ LDS. This prevents the LDS from running independently, and thereby hinders parallelization.

**Related Work** There is a vast literature on system identification [Ljung, 1999]. Subspace identification (SSID) is the prevalent technique, utilized by the state-of-the-art work cited in the introduction. SSID does not reduce MIMO to SIMO, as we do. It is well known that the controllability indices are numerically unstable [Jordan and Sridhar, 1973]. Our result shows this numerical instability is a blessing, since a small perturbation renders it useful. There are deterministic methods of modifying the original system to obtain (nearly) equal controllability indices, at the expense of increased state size [Cook, 1978]. The (mis)use of MIMO canonical forms as parameterizations for learning is discussed in [Glover and Willems, 1974]. They discuss a numerical advantage of Luenberger's (pseudocanonical) form over MIMO canonical forms, and base a system identification method upon it [Glover, 1973]. Subsequent works on 'overlapping' parameterizations also avoided the problem of unknown structural indices [Corrêa and Glover, 1984, Gevers and Ah-Chung, 1985].

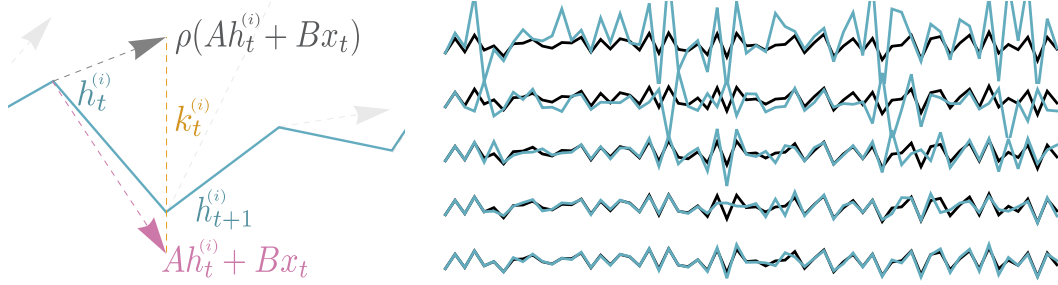

Figure 4: *Left*: Visualization of the local corrections within LDStack. Suppose the $i$th layer's states $h_t^{(i)}$ are all computed. We consider, at each $t$, two hypothetical steps from $h_t^{(i)}$: the linear step $Ah_t^{(i)} + Bx_t$ and the nonlinear step $\rho(Ah_t^{(i)} + Bx_t)$. Their difference is the correction $k_t^{(i)}$, which is added to $h_t^{(i+1)}$ in the next layer. Note that $h_{t+1}^{(i+1)} = Ah_t^{(i)} + Bx_t + k_t^{(i-1)}$ does not coincide with the hypothetical linear step, since it was corrected in a similar manner. The faint gray arrows illustrate that the corrections are computed in parallel using only local information. *Right*: RNN (black) approximated using stacked LDS of increasing depth (from top to bottom). Observe the "correct from the start" behavior described in Proposition 5.

## 5 Approximating Nonlinear RNNs by Stacked LDS

Let $h_{t+1} = \rho(Ah_t + Bx_t)$ be an RNN which takes inputs $x_t \in \mathbb{R}^d$ and an initial state $h_0 \in \mathbb{R}^n$, and produces subsequent states $h_t \in \mathbb{R}^n$. Its nonlinearity $\rho$ has deviation from linearity $\delta(a) = \rho(a) - a$. This deviation is used to define local corrections to an LDS, as follows:

$$h_{t+1} = (Ah_t + Bx_t) + \delta(Ah_t + Bx_t) \longrightarrow h_{t+1}^{(i+1)} = Ah_t^{(i+1)} + Bx_t + \overbrace{\delta(Ah_t^{(i)} + Bx_t)}^{k_t^{(i)}} \quad (6)$$

On the left is a trivial equality involving $\delta$. Its first term is a linear transition from $h_t$; its deviation from a correct (nonlinear) transition is measured by the second term. The approximation starts with a plain LDS $h_{t+1}^{(0)} = Ah_t^{(0)} + Bx_t$; then, its deviations are used as corrections $k_t^{(0)}$ to a subsequent LDS. Iterating this construction yields a stack of corrected LDS. As the previous layer's states $h_t^{(i)}$ become close to the next layer's $h_t^{(i+1)}$, the corrections become more accurate. With enough layers, the nonlinear RNN is exactly recovered. More generally, the layers are "correct from the start". Since the initial state $h_0^{(0)} = h_0$ is correct, the first layer gets the first state correct: $h_1^{(1)} = Ah_0 + \delta(Ah_0) = h_1$. The second layer gets the second state correct, and so forth, yielding a consistency guarantee.

**Proposition 5.** $h_t^{(\Delta)} = h_t$ *for all* $t \in [\Delta]$. *Thus,* $h^{(T)} = h$.

Since the stacked LDS have nonlinearity along depth, they may seem just as difficult to analyze as the original nonlinear RNN. Fortunately, our construction is a discrete, additive version of a continuous, multiplicative scheme developed in control theory [Tomás-Rodríguez and Banks, 2010]. It has been extensively used to analyze nonlinear dynamical systems via sequences of linear approximations. Controllers for aircraft, supertankers, and autopilots have been derived with this approach [Çimen and Banks, 2004]. It is possible to derive explicit solutions for the linear approximation in terms of an underlying Lie algebra [Banks, 2002]. The appendix describes this control-theoretic precursor of our construction. It is reasonable to expect that some of the same analytic techniques will carry over.

**Related Work**. Generalizing earlier works [Balduzzi and Ghifary, 2016, Bradbury et al., 2017], Martin and Cundy [2018] advocate the removal of nonlinearities across time, while introducing nonlinearity along depth. Given an RNN, they replace nonlinear dependencies across time with a "linear surrogate" amenable to PLR. These new RNNs can run in parallel, but it is not clear they can approximate the original nonlinear RNNs, and they are not as well-studied as LDS. Restricted subclasses of RNNs can be approximately differentiated in constant time [Liao et al., 2018]. There are substantial efforts to understand nonlinear RNNs [Karpathy et al., 2015] and develop provable learning algorithms for them [Allen-Zhu and Li, 2019, Allen-Zhu et al., 2019, Foster et al., 2020].

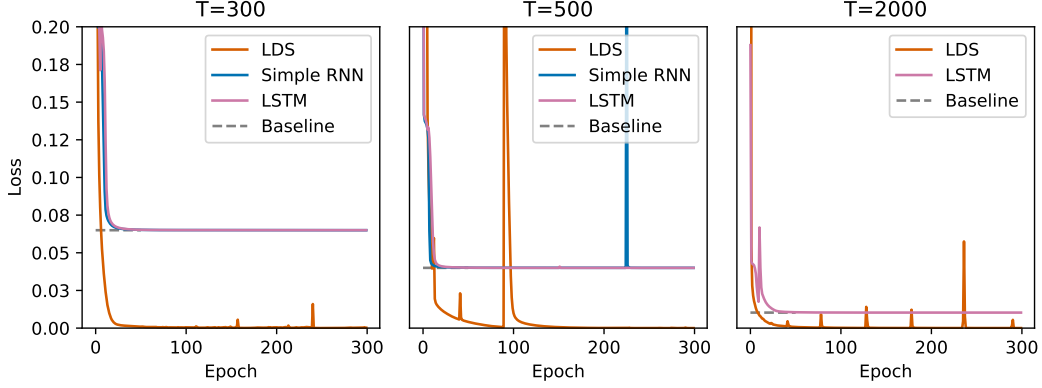

Figure 5: On the copying memory problem, standard RNNs do not outperform a trivial baseline. We solve it with the simplest model to date: a unitary SISO LDS, as described in Figure 1.

The culmination of our results is the neural network layer $\mathbf{LDStack}(\boldsymbol{\rho}, \boldsymbol{n}, \boldsymbol{\Delta}, \boldsymbol{r})(\boldsymbol{x}, \boldsymbol{h_0})$. It takes (a batch $x$ of) length-$T$ sequences of $d$-dimensional vectors, and an $n$-dimensional initial state $h_0$. It returns (a batch $\hat{h}$ of) length-$T$ sequences of $n$-dimensional states. It uses $O(\Delta n^2 \log T \log r)$ time on $O(rT)$ parallel processors.[2] Its settings are the nonlinearity $\rho$, state size $n > 1$, depth $\Delta \geq 1$, and number of projections $r \geq 1$. It has $O(n + n^2 d)$ trainable weights and $rd$ fixed weights.

**LDStack details**. Suppose the (unknown) RNN has parameters $(\tilde{A}, \tilde{B})$ which define a reachable LDS. Let $\mathcal{C}$ be its $n \times (n \times d)$ controllability matrix. At initialization, random projections $g_j \in \mathbb{R}^d$ are drawn, for $j \in [r]$. The first layer is an average of plain SIMO LDS. Let $x_t^{[j]} = x_t^T g_j$ be the projected input of the $j$th SIMO LDS. $s_{j,t+1}^{(0)'} = \lambda \circ s_{j,t}^{(0)'} + B' x_t^{[j]}$ are computed in parallel, per Section 3. To compute the corrections, reverse the canonical and diagonal transformations $\mathcal{T}_j$ and $V$ according to (2) and (3). Recall from (2) that $\mathcal{T}_j^{-1} = \mathcal{C}_j$, the $n \times n$ controllability matrix of the $j$th SIMO LDS $(\tilde{A}, \tilde{B}g_j)$. Then $\mathcal{C}_j = [Bg_j, ABg_j, \ldots, A^{n-1}Bg_j] = \mathcal{C} \cdot g_j$. Eliding superscripts: $s'_{j,t}$

$$\tilde{A}\tilde{s}_{j,t} + \tilde{B}x_t = \mathcal{T}_j^{-1}A\underbrace{\mathcal{T}_j\tilde{s}_{j,t}}_{s_{j,t}} + \mathcal{T}_j^{-1}Bx_t = \mathcal{T}_j^{-1}(As_{j,t} + Bx_t) = \mathcal{T}_j^{-1}V^{-1}(\Lambda \overbrace{Vs_{j,t}} + VBx_t)$$
$$= \mathcal{T}_j^{-1}V^{-1}(\lambda \circ s'_{j,t} + B'x_t)$$

We introduce a free parameter $W \in \mathbb{C}^{n \times n \times d}$ which ideally satisfies $W \cdot r_j = (\mathcal{C} \cdot r_j)V^{-1}$, so it can directly perform the reverse transformations $\mathcal{T}_j^{-1}V^{-1}$. Averaging within (6), the corrections are $\tilde{k}_t^{(0)} = \delta(\frac{1}{r}\sum_{j=1}^r \tilde{A}\tilde{s}_{j,t}^{(0)} + \tilde{B}x_t^{[j]})$. Now we compute the next layer. Take the corrections back to the diagonalized basis as $k_{j,t}^{(0)'} = V\mathcal{T}_j\tilde{k}_t^{(0)}$. The corrected SIMO LDS are run in parallel using $s_{j,t+1}^{(1)'} = \lambda \circ s_{j,t}^{(1)'} + B'x_t^{[j]} + k_{j,t}^{(0)'}$. After $\Delta$ layers, $\hat{h}_t^{(\Delta-1)} = \frac{1}{r}\sum_{j=1}^r \tilde{s}_{j,t}^{(\Delta-1)}$ are returned.

## 6 Experiments

**Copy memory problem** [Arjovsky et al., 2016, Hochreiter and Schmidhuber, 1997]. The goal is to remember the first 10 entries $r$ of an input sequence, withhold output for $T$ steps (for which the inputs are just "blanks"), and, upon seeing a "go" input at time $T + 10$, to output $r$. There is a SISO LDS which achieves zero error [Henaff et al., 2016], so we do not consider LDStack of higher depth. Unitary RNNs are known to solve the problem, so we use the unit parameterization of Figure 1. Arjovsky et al. [2016] use LSTM, simple tanh RNN, and uRNN of respective sizes $n = 40, 80$, and $128$ for parameter counts of roughly 6500. We use $n = 160$, which results in just 3380 parameters, including $C' \in \mathbb{C}^{n \times n}$. Our solution is the state of the art: it uses the simplest (linear) RNN with the fewest parameters to solve the $T = 2000$ instance. This has demanded full-capacity uRNNs [Wisdom et al., 2016] or subsequent nonlinear RNNs [Lezcano-Casado and Martínez-Rubio, 2019].

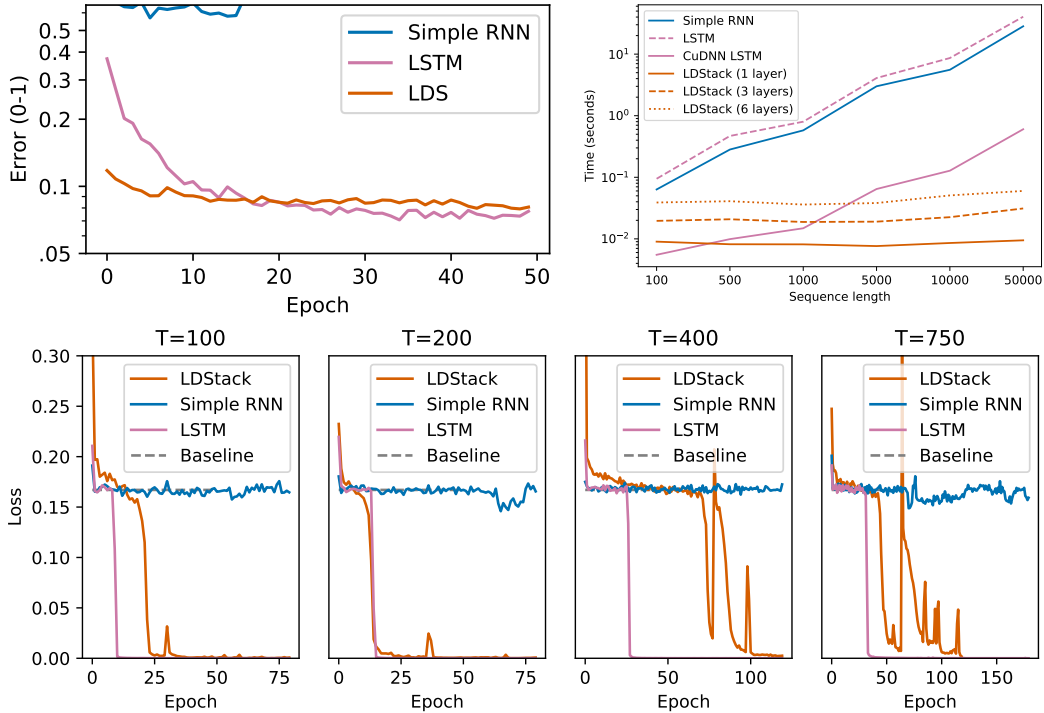

Figure 6: *Top left*: Sequential permuted MNIST. *Top right*: Runtimes for different sequence lengths. *Bottom*: the adding problem, with larger sequence lengths representing more challenging problems.

**Sequential permuted MNIST**. The images are presented as length-784 sequences of pixels. Their order is arbitrary, but fixed across all images. We compare an $n = 384$ SIMO LDS having ~16,500 parameters to an $n = 128$ LSTM having ~68,000 parameters, as well as an $n = 128$ tanh RNN having ~18,000. The LDS and the LSTM achieve similar accuracies of 91.8% and 92.3%. This performance is not state of the art: for example, Chang et al. [2018] achieve 95.8% accuracy with 10,000 parameters. However, the LDS steps take 73ms, compared to 324ms for the unfused RNN.

**Runtime comparison**. LDStack (prototype code in both Python and CUDA) is always faster than unfused RNNs. At longer sequence lengths, it is even faster than the highly-optimized, fused CuDNN LSTM. The $O(T)$ and $O(\log T)$ asymptotics manifest plainly.

**Adding problem** [Arjovsky et al., 2016, Hochreiter and Schmidhuber, 1997]. Each input has dimension $T \times 2$. The output is the sum of the two numbers (from the first dimension) which are marked by ones (in the second dimension); the rest of the numbers are marked by zeros. Trivially returning 1 achieves mean-squared error 0.167. This problem cannot be solved by an LDS, so it exercises both random projection and nonlinear approximation by stacking. We use LDStack with state size $n = 32$, depth $\Delta = 2$, and $r = 6$ projections. This has 4,175 parameters, compared to ~27,000 and ~17,000 for an LSTM and tanh RNN, respectively, having $n = 80$. The simple RNN fails to beat the trivial baseline. The LSTM and LDStack both solve the problem up to $T = 750$, though the latter takes longer to converge, and is more unstable in later epochs.

## 7   Conclusion and Future Work

This paper presents a new program for developing fast and trustworthy RNNs, based on the core primitive of SIMO LDS. In order for this program to succeed, significant limitations must still be overcome. Approximation guarantees for low-depth stacks must be studied. Although LDStack scales well with $T$, it is inefficient in other respects: memory use scales with depth, and parameters scale as $O(n^2d)$. We have not closely examined algorithms for learning LDStack, even though RNNs suffer from the vanishing/exploding gradient problem. Finally, deep learning primitives are heavily optimized for GPUs [Chetlur et al., 2014]; our implementation requires similar treatment.

## 8 Broader Impact

The broad impact of our work is to make RNNs faster and more trustworthy. Trustworthiness - encompassing the topics of robustness, interpretability, and fairness - is a major concern about deep learning. In many applications, trustworthiness is as important as the traditional metrics of speed and accuracy. Lack of trust is now hindering adoption of machine learning in healthcare, law, social media, and other fields. In this work, we hope to bolster society's faith in machine learning models, particularly recurrent neural networks, without sacrificing the speed and accuracy which are also required of them. Responsible applications of our work will balance trustworthiness, speed, and accuracy according to the best interests of those affected by the resulting algorithm.

## 9 Funding Disclosure

Additional revenues related to this work: paid talks and work at Allergan plc and Estee Advisors.

## Footnotes

[1] In continuous time, reachability and controllability are equivalent. In discrete time, they are equivalent when $A$ is nonsingular.

[2] For simplicity, this time bound does not internally parallelize $O(n^2)$ matrix-vector multiplication and linear system solving. The analogous bound for nonlinear RNNs is $O(n^2 T)$.

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
