[Supplementary Material]

# 10 Appendix

## 10.1 Proof of Lemma 2

*Proof.* We wish to show $Au_j = \lambda_j u_j$. If the theorem is true, then $\lambda_j u_{j,i} = \lambda_j \frac{1}{\lambda_j^{n-i}} = \frac{1}{\lambda_j^{n-(i+1)}} = u_{j,i+1}$. Recall the state update of the controllable LDS, which shifts $n-1$ entries and computes a dot product in the last entry:

$$Au_j = \begin{bmatrix} u_{j,2} \\ \vdots \\ u_{j,n-1} \\ -\sum_i a_{i-1} u_{j,i} \end{bmatrix} = \begin{bmatrix} \lambda_j u_{j,1} \\ \vdots \\ \lambda_j u_{j,n} \\ -\sum_i a_{i-1}/\lambda_j^{n-i} \end{bmatrix}$$

It suffices to show:

$$-\sum_i a_{i-1}/\lambda_j^{n-i} = \lambda_j u_{j,n} = \lambda_j \text{ i.e. } \sum_{1 \le i \le n} \frac{a_{i-1}}{\lambda_j^{n-(i-1)}} = -1 \tag{7}$$

It is well known that the characteristic polynomial of $A$ is $p(t) = a_0 + a_1 t + a_2 t^2 + \ldots + a_{n-1} t^{n-1} + t^n$. By definition, its roots (those $t$ where $p(t) = 0$) are the eigenvalues of $A$.
So each $\lambda_j$ satisfies:

$$0 = a_0 + a_1 \lambda_j + a_2 \lambda_j^2 + \ldots + a_{n-1} \lambda_j^{n-1} + \lambda_j^n = \lambda_j^n \left( 1 + \sum_{1 \le i \le n} \frac{a_{i-1}}{\lambda_j^{n-(i-1)}} \right)$$

Either we have a null eigenvalue $\lambda_j = 0$, or we have the desired equation (7). $\square$

## 10.2 Proof of Lemma 3

*Proof.* Let $v_i$ be the $i$th row of $U^{-1}$. The dual basis of $U$ is $(U^{-1})^T$, i.e. $u_i^T v_i = 1$ and for all $j \ne i$, $u_i^T v_j = 0$. Since $B'$ is the conjugate of the $n$th column of $U^{-1}$, it is determined by the $n$th coordinates of the $v_i$. We derive these by employing the adjugate technique of Denton et al. [2019]. Recall the determinant $\det(A) = \prod_i \lambda_i$ is the product of the eigenvalues. Also recall the following general definition of the adjugate matrix, when $A$ is diagonalizable but not necessarily Hermitian:

$$\text{adj}(A)_{i,j} = \sum_{k=1}^n \left( \prod_{l \ne k} \lambda_l \right) u_{k,i} \bar{v}_{k,j}$$

For any $k$, replace $A$ by $\lambda_k \lambda I_n - A$. This causes all but one of the summands to vanish, yielding the following simplication:

$$\text{adj}(\lambda_k I - A)_{i,j} = \left( \prod_{l \ne k} (\lambda_k - \lambda_l) \right) u_{k,i} \bar{v}_{k,j}$$

Setting $i = 1$ and $j = n$, and substituting the previously derived entries of $u_k$:

$$\text{adj}(\lambda_k I - A)_{1,n} = \left( \prod_{l \ne k} (\lambda_k - \lambda_l) \right) \frac{1}{\lambda_k^{n-1}} \bar{v}_{k,n} \tag{8}$$

By the Laplace expansion of the adjugate matrix of $A$, $\text{adj}(\lambda_k I - A)_{1,n} = (-1)^{1+n} \det(M)$, where $M$ is the minor of $\lambda_k I - A$ produced by removing its $n$th row and 1st column. It is straightforward to show that the only eigenvalue of $M$ is $-1$ with multiplicity $n-1$, and therefore $\det(M) = (-1)^{n-1}$. Therefore $\text{adj}(\lambda_k I - A)_{1,n} = (-1)^{2n} = 1$. Combining this with (8) obtains an equality for each $\bar{v}_{k,n}$, which matches the desired result. $\square$

## 10.3 Proof of Proposition 3

Proposition 3 is an easy corollary of the following proposition, which involves MISO LDS rather than MIMO LDS.

**Proposition 6.** *Let $x_1, \ldots, x_T$ be any sequence of $d$-dimensional inputs, and let $y_1, \ldots, y_T$ be the corresponding outputs of a reachable MISO LDS with parameters $(A, B, C, D)$. For each $j \in [r]$, let $g_j$ be a $d$-dimensional standard normal vector, $x_t^{[j]} = g_j^T x_t$ be a projected sequence of scalar inputs, and $(A, Bg_j, C, D)$ be the parameters of a SISO LDS producing outputs $y_t^{[j]}$. Let $\hat{y}_t = \frac{1}{r} \sum_{j=1}^r y_t^{[j]}$ be the average output. For each $t \leq T$, $\mathbf{E}(y_t - \hat{y}_t)^2 = 2 \left\| Z_t \right\|_F^2 / r$, where $Z_t$ is defined below in (9). Furthermore, the SISO LDS are almost surely reachable, and share the same canonical form matrix.*

*Proof.* While proving this result, let us take $D = 0$ and $s_0 = 0$ for notational simplicity. (These are just constant terms which do not affect the result.) From the convolution representation (1) and the random construction of the SISO LDS, we find that the approximation is unbiased:

$$\mathbf{E}\, \hat{y}_t = \mathbf{E}\, \frac{1}{r} \sum_j \sum_{\tau=1}^{t-1} CA^\tau Bg_j g_j^T x_\tau = \sum_{\tau=1}^{t-1} CA^\tau B \left( \frac{1}{r} \mathbf{E} g_j g_j^T \right) x_{t-\tau} = y_t$$

Therefore the mean squared error is just the variance:

$$\mathbf{E}\, (y_t - \hat{y}_t)^2 = \mathbf{E}\, ((\mathbf{E}\, \hat{y}_t) - \hat{y}_t)^2 = \mathbf{V}(\hat{y}_t)$$

By the independence of the $g_j$, and the cyclic property and linearity of trace, we reduce to the variance of a quadratic in normal variables:

$$
\begin{aligned}
\mathbf{V}(\hat{y}_t) =&\, \mathbf{V} \left( \sum_{\tau=1}^{t-1} \mathrm{tr}(CA^\tau B \left( \frac{1}{r} \sum_{j=1}^r g_j g_j^T \right) x_{t-\tau}) \right) \\
=&\, \frac{1}{r^2} \sum_{j=1}^r \mathbf{V} \left( \sum_{\tau=1}^{t-1} \mathrm{tr}(g_j^T x_{t-\tau} CA^\tau B g_j) \right) \\
=&\, \frac{1}{r^2} \sum_{j=1}^r \mathbf{V} \left( g_j^T g_j \sum_{\tau=1}^{t-1} CA^\tau B x_{t-\tau} \right) \\
=&\, \frac{1}{r^2} \sum_{j=1}^r \mathbf{V} \left( g_j^T \underbrace{\sum_{\tau=1}^{t-1} x_{t-\tau} CA^\tau B}_{Z_t} g_j \right)
\end{aligned}
\tag{9}
$$

The inner quadratic is not changed by replacing $Z_t$, which is asymmetric, with $\bar{Z}_t = \frac{1}{2}(Z_t + Z_t^T)$, which is symmetric, diagonalizable, and shares the same eigenvalues $\nu_1, \ldots, \nu_d$. $g_j$ retains its distribution under the rotation $U$ that diagonalizes $\bar{Z}_t$. We find the variance is just the squared Frobenius norm of $Z_t$:

$$
\begin{aligned}
\mathbf{V} \left( g_j^T \bar{Z}_t g_j^T \right) =&\, \mathbf{V} \left( g_j^T U^T \mathrm{diag}(\nu) U g_j \right) \\
=&\, \mathbf{V} \left( \sum_{i=1}^d g_{j,i}^2 \nu_i \right) = 2 \sum_{i=1}^d \nu_i^2 = 2 \left\| Z_t \right\|_F^2
\end{aligned}
$$

Now we verify that the SISO LDS are almost surely reachable, assuming the MISO LDS is reachable. By Lemma 1, we must show that if $[\gamma I - A; B]$ has full rank for all $\gamma \in \mathbb{C}$, then $[\gamma I - A; Bg_j]$ also does, almost surely. This holds because $g_j$ has density with respect to Lebesgue measure.

To conclude the proof of Proposition 6, denote the MIMO LDS matrices above as $(\tilde{A}, \tilde{B})$. When projected to SIMO LDS $(\tilde{A}, \tilde{B}g_j)$, their canonical forms $(A_j, B)$ are obtained via $\tilde{A}_j = \mathcal{T}_j^{-1} A \mathcal{T}_j$. Let $v_i$ and $\lambda_i$ be an eigenvector and corresponding eigenvalue of $\tilde{A}$: $\tilde{A} v_i = \lambda_i v_i$. Then $A_j \mathcal{T}_j v_i = \lambda_i \mathcal{T}_j v_i$, so the $A_j$ share the same eigenvalues as $\tilde{A}$. Since $A_j$ are companion matrices of the same form (2), this means they are actually the same matrix $A$.

$\square$

## 10.4 Proof of Proposition 4

The following proposition implies Proposition 4.

**Proposition 7.** *Let $n$ be divisible by $d$. Let $A \in \mathbb{R}^{n \times n}$ and $B \in \mathbb{R}^{n \times d}$ be full rank. Let $(A, B)$ form a reachable MIMO LDS. Choose any $\epsilon > 0$ and any (Schatten) matrix norm $||\cdot||$. There is a $\delta > 0$ such that the following holds. Let $G$ be an $n \times n$ matrix of normal variables of mean zero and variance $\delta$, and $\tilde{A} = A + G$. Then, with nonzero probability, $\left|\left| A - \tilde{A} \right|\right| \leq \epsilon$ and the controllability indices of $(\tilde{A}, B)$ are all equal to $n/d$.*

*Proof.* Clearly $||G|| \leq \epsilon$ with nonzero probability. The controllability indices are equal if the first $n$ rows of the controllability matrix (4) are linearly independent. Thus, we must show that the following $n \times n$ matrix has full rank:

$$\mathcal{C}_{:,:n} = [B, (A + G)B, (A + G)^2 B, \dots, (A + G)^{n/d-1} B]$$

The first $d$ columns are linearly independent by assumption. In the remaining columns, since $G$ is normal — and therefore has density with respect to Lebesgue measure — linear independence follows from a standard argument. $\mathcal{C}_{:,:n}$ is full rank unless its determinant is zero. The determinant is a polynomial $p : \mathbb{R}^{n^2} \to \mathbb{R}$ in the (flattened) entries of $\mathcal{C}_{:,:n}$. For any such polynomial $p$, the set $p = 0$ has Lebesgue measure zero. $\square$

## 10.5 Approximation of Nonlinear Systems by Time-Varying LDS

Tomás-Rodríguez and Banks [2010] describe a method of approximating continuous-time dynamical systems by linear, time-varying ones. We briefly review their method, showing how it gives rise to a multiplicative variant of LDStack. Consider the following nonlinear, discrete-time dynamical system: $h_{t+1} = \rho(Ah_t) + Bx_t$. $Bx_t$ is usually inside the nonlinearity $\rho$, but we keep it separate for reasons that will be discussed below. $\rho$ must be continuously differentiable. Furthermore, in order for the approximation scheme to be numerically stable, $\rho$ must also be analytically "nice", as described below. We use the inverse square root activation $\rho(a) = a/\sqrt{1 + a^2}$ as a running example.

We begin by viewing the RNN as an Euler discretization of a continuous-time dynamical system (e.g. Tallec and Ollivier [2018]). Using the Taylor expansion $h(t + \epsilon t) \approx h(t) + \epsilon t \cdot \dot{h}(t)$, and taking a step size of $\epsilon = 1$, we obtain the following nonlinear differential equation: $\dot{h} = \rho(Ah) - h + Bx$. (We elide the dependence on $t$ to simplify notation). The first step is to convert the dynamical system to state-dependent coefficient (SDC) form: $\dot{h} = \mathcal{A}(h)h - h + Bx$. Here, the nonlinear update is factorized to resemble an LDS. SDC form does not allow $\mathcal{A}$ to depend on $x$, which is why $Bx_t$ was kept outside of $\rho(\cdot)$. The SDC factorization can be derived in a straightforward manner.

**Lemma 4.** *The following is a valid SDC factorization when $\rho \in C^1$ and $\rho(0) = 0$. [Cimen, 2010]*

$$\mathcal{A}(h) = \int_0^1 \left. \frac{d\rho(Ah)}{dh} \right|_{h=\lambda h} d\lambda$$

We call $\rho$ "nice" if the above factorization is numerically stable and can be analytically derived. For our example $\rho$, a brief calculation shows the SDC form is:

$$\dot{h} = \underbrace{\text{diag}(1/\sqrt{1 + (Ah)^2})A}_{\mathcal{A}(h)} h - h + Bx$$

Note that $\mathcal{A}(h)h$ is a multiplicative, entrywise correction of $Ah$ based on its deviation from $\rho(Ah)$. Under weak conditions on $\mathcal{A}$, the SDC-form nonlinear system can be approximated by a sequence of linear, time-varying systems.

**Theorem 1** (Informal). *Let $\mathcal{A}$ be locally Lipschitz. Consider this sequence of time-varying LDS:*

$$\dot{h}^{(0)} = \mathcal{A}(h_0)h^{(0)} - h^{(0)} + Bx \qquad\qquad h_0^{(0)} = h_0$$
$$\dot{h}^{(i)} = \mathcal{A}(h^{(i-1)})h^{(i)} - h^{(i)} + Bx \qquad\qquad h_0^{(i)} = h_0$$

*As $i \to \infty$, the solution of $h^{(i)}$ converges to the solution of $h$. [Tomás-Rodríguez and Banks, 2010]*

Figure 7: Additive and multiplicative approximations of a nonlinear RNN (black). The latter converge more quickly than the former, at least when the same matrix $A$ is shared among the nonlinear RNN and the approximating LDS.

The nonlinear RNN approximation in Definition 2 is just a discretization of Theorem 1.

**Definition 2** (Nonlinear RNN Approximation). *Let $\rho$ be a continuously differentiable activation function with $\rho(0) = 0$. For $t \in [T]$, let $h_{t+1} = \rho(Ah_t) + Bx_t$ be the $n$-dimensional states of an RNN with parameters $(A, B)$. Let $\mathcal{A} : \mathbb{R}^n \to \mathbb{R}^{n \times n}$, as given by (4), be locally Lipschitz. This is a stack of time-varying LDS whose depth is indexed by $i$:*

$$
\begin{aligned}
h_{t+1}^{(0)} &= \mathcal{A}(h_0)h_t^{(0)} + Bx_t & h_0^{(0)} &= h_0 \\
h_{t+1}^{(i)} &= \mathcal{A}(h_t^{(i-1)})h_t^{(i)} + Bx_t & h_0^{(i)} &= h_0
\end{aligned}
$$

Our additive variant is more algorithmically convenient, whereas the multiplicative variant is superior for approximation theory. Multiplicative corrections interfere with diagonalization, which is crucial for our algorithms. However, as illustrated in Figure 7, additive corrections can produce oscillations which lead to slower convergence. Note that this occurs when the LDS matrix $A$ matches that of the nonlinear RNN - a choice made for analytic simplicity, when $A$ is known. At relatively small depths $\Delta$, it may be possible to achieve better approximation with a different LDS matrix $A_\Delta$. In a practical learning setting, $A_\Delta$ is learned directly, without any reference to the unknown $A$.

### 10.5.1  Another Eigenvalue Parameterization

A problem with the standard $(\alpha, \beta)$ parameterization of $\lambda$ is that the number of real and complex eigenvalues is hardcoded. Two real eigenvalues cannot "cross over" to being complex conjugate pairs, and vice versa. To remedy this, we might consider independently parameterizing the real and imaginary parts of $\lambda$ with $2n$ reals. Unfortunately, this does not constrain the complex numbers to be conjugate pairs, so then $\lambda_1, \ldots, \lambda_n$ are not necessarily the eigenvalues of a real matrix $A$. The following "hinge" parameterization, defined in terms of two real numbers $(\alpha, \omega)$, avoids both of these issues. Let $h(a) = \max(0, a)$ be a ReLU. Consider these values:

$$\alpha + h(-\omega)i \qquad \text{and} \qquad \alpha + h(\omega) - h(-\omega)i$$

If $\omega > 0$, then the values simplify to $\alpha$ and $\alpha + \omega$, which are real. If $\omega < 0$, they simplify to $\alpha \pm \omega i$, which are complex conjugate pairs. The values are distinct when $\omega \neq 0$.

### 10.6  Additional Experiment Details

In all the experiments, we used Adamax [Kingma and Ba, 2014] as the optimizer for LDS and LDStack. In some situations, we observed this choice substantially improved the rate of convergence. We used Adam as the optimizer for the LSTM and simple RNN. Abbreviate the learning rate and batch size as $\eta$ and $B$, respectively. For the copy memory problem, $\eta = 0.01, B = 256$. For the runtime comparison, $n = 32$ and $B = 4$. For sequential permuted MNIST, $B = 128$. LDS used $\eta = 0.0003$, and the hinge parameterization described in Section 10.5.1. LSTM and simple RNN used $\eta = 0.01$. In the adding problem, $B = 32$ there were 100 steps per epoch. LDStack used $\eta = 0.003$ and the hinge parameterization. We observed faster convergence with a smaller $n = 32$ model LDStack than with a larger $n = 64$ one. LSTM and simple RNN used $\eta = 0.01$.