[Reviews · NeurIPS 2020]

Review 1

Summary and Contributions: The paper proposes to use Single-input, multiple-output linear dynamical systems as a building block for complex RNNs. The author shows that reachable SIMO LDS can be computed in parallel. A theoretical analysis is provided on the conditioned MIMO LDS is arbitrarily close to a coupling of d SIMO LDS and a simpler method is presented to approximately implement it. The new method is encapsulate time-varying LDS by LDSTACK to approximate nonlinear RNN.

Strengths: The strengths of this work including propose a theoretical analysis on using LDS in RNN, a new method to approximate nonlinear RNN with LDS, a theoretical analysis on the relation between SIMO LDS and MIMO LDS, a scheme for approximate MIMO LDS with SIMO LDS.

Weaknesses: This paper seems to be a paper with two style, well-written in section 1 to section 4 and not good enough in section 5 and section 6. Section 5 and section 6 seems to be quite strange to this paper. First, as the culmination of this paper, the analyze and the formulation about LDStack in section 5 is not clear enough, which makes this paper confusing, Second, the experimental result in section 6 is not clear enough. The author describe the background of the experiment, but something is confusing about section 6. The relation between section 6 and previous section is not clear enough. Besides, there are some small mistakes, e.g., Figure 6 is an empty figure.

Correctness: The adopted methodology seems to be correct. The empirical results seems to be convinced.

Clarity: Satisfactory, but not good enough in supplementary and experiments.

Relation to Prior Work: Yes.

Reproducibility: Yes

Additional Feedback:


Review 2

Summary and Contributions: The paper proposes a new algorithm for learning sequence to sequence tasks such as those in time-series and language. The algorithm relies on a single-input linear dynamical system (LDS) as a building block where multiple LDSs are coupled to handle multiple inputs and then stacked together to become functionally equivalent to a recurrent neural network (RNN). Different from an RNN, the proposed stacked-LDS can be parallelized and is easier to analyze. The authors start with a theoretical foundation of the proposed model with detailed proofs and they end by comparing their model to RNNs and apply it to real data.

Strengths: While powerful, RNNs are known to be both computationally expensive and difficult to train. This attracted the community to other sequence models such as convolutional or attention-based. This paper introduces a novel algorithm using linear dynamical systems (LDS) as another alternative to RNNs. They show that their algorithm can be parallelized and since it is based on linear components it can be easily analyzed and explained. This can be a very attractive method to further our understanding of sequence models and RNNs specifically. Both the theory and results presented in this paper were impressive.

Weaknesses: When comparing LDStack to RNNs a number of concerns arise: - As mentioned by the authors, the dynamics of the RNN are viewed as an Euler discretization and the nonlinear update rule is essentially linearized to work with LDS. Do we understand the limitations of this approximation? Furthermore, wouldn't we get an accumulation of the error over time during inference? - One advantage of RNNs over other sequence models is their ability to process long term memory, can LDS retain that property? or do the different stacks have to operate on different time-scales?

Correctness: Both the theoretical assumptions, proofs and results were described in detail and are well-explained with no visible errors.

Clarity: The paper is well written, detailed mathematical derivations were well presented as well as their proofs. Experimental results are explained well in the figures.

Relation to Prior Work: The authors cite various contributions from various fields including control theory, dynamical systems and machine learning. The novelty and differences from other studies is well described.

Reproducibility: Yes

Additional Feedback:


Review 3

Summary and Contributions: In this paper, it is shown that for reachable SIMO LDS, i.e. single-input multi-output (time-invariant) linear dynamical systems, the output, and its gradient w.r.t. parameters can be efficiently computed due to the Parallel Linear Recurrence Algorithm introduced in previous works. This potentially leads to efficient training for RNNs with long sequence data. The result applies to training more complicated model in the following ways: 1. For MIMO (multi-input multi-output) LDS, given suitable condition, a perturbed system, which can be regarded as multiple coupled SISO LDS, serves as a good approximation; 2. For MIMO LDS, consider multiple SIMO LDS by randomly projecting the original input onto 1-dimension subspaces, the averaged output of these SIMO LDS can approximate the output of the original system given sufficiently large number of projections; 3. For nonliear RNNs, nonliear operation from layer to layer is approximated by a stack of time-varying LDS.

Strengths: This paper suggested several approaches to approximate complicate dynamical systems by combination of simple linear systems, the claims and propositions regarding the approximation accuracy are well developed and supported by the numerical simulations. In the broad sense, these approaches are examples of identifying complicated systems by combination of simple systems with reasonable accuracy, which, as claimed by the author, might improve the computation efficiency.

Weaknesses: To well motivate the approach of system identification with combination of linear systems. The LDStack, the RNN structure proposed in the paper, is supposed to: 1) have as rich expressive power as ordinary RNNs; 2) have some advantages over ordinary RNNs in some aspects such as computational complexity, dimension of parameter space, etc. In this paper, 1) is theoretically and numerically supported, while 2) is not seen from the numerical experiments. To be specific, one of the motivation of such approach is the high computational efficiency of training SIMO LDS for long sequence data. However, in the experiments there is no comparison between LDStack and ordinary RNNs regarding computation efficiency, possibly with dependency on the length of the sequence data.

Correctness: yes

Clarity: Theoretical part: well written with a few minor typos and problems Algorithm part: Overall good but would be better with more details on the implementation of the LDStack: explicit parameterization and the gradients. Experiment: see Weaknesses.

Relation to Prior Work: yes

Reproducibility: Yes

Additional Feedback: typos: 1. line 109, C is missing in the canonical form; e_n, e_2 shoud be e_n(\lambda), e_2(\lambda). 2. line 124 Lemma 2, A is required to have the canonical form. The section 2.2 didn't explicitly specify that. 3. Typos in Figure 1. the state transition and output equations. Comments: 1. line 170 Lemma 3, there is the degenerate case where A has repeated eigenvalues, which is not considered here (actually the reachablity prevents this case, but it is possible to have two eigenvalues close to each other, does that causes any numerical issues?) 2. line 461 in supplement, the proof only consider the case when A is normal, which is insufficient.

[Author Response · NeurIPS 2020]



We thank all the reviewers for their careful attention to our paper, and their helpful comments.

**Speed**. See above Figure 1 (top left) for runtime comparison among RNNs. LDSTACK (research code in both Python
and CUDA) is always faster than unoptimized RNNs. At longer sequence lengths, it is even faster than the highly-
optimized, fused CuDNN LSTM. We didn't initially report this comparison because we will release an improved
LDSTACK implementation, and expect it will be faster than CuDNN LSTM at essentially all problem sizes. Currently,
LDSTACK begins with an expensive transpose, since our CUDA op takes time-major sequences. We use homemade
(unoptimized) prefix scans and reductions, rather than those in the CUB library. Finally, the LDSTACK op can be fused.

**Exposition in Sections 5 and 6**. LDSTACK has a simple intuition, summarized as follows. The first layer is a plain
LDS. Subsequently, at layer $i$, the previous layer's states $s^{(i-1)}$ are used to estimate where linear transitions incorrectly
deviate from nonlinear ones. This is measured by $p_t^{(i-1)} = \delta(As_t^{(i-1)})$ where $\delta(a) = \rho(a)/a$ is the (multiplicative)
deviation of the RNN's nonlinearity $\rho$. These correct the transitions within layer $i$ via $s_{t+1}^{(i)} = \text{diag}(p_t^{(i-1)})As_t^{(i)} + Bx_t$.
We regrettably removed this simpler explanation to fit the 8-page limit. We kept the current (rather abstract) exposition
because we did not want to obscure the origins of the idea of stacking LDS. This was part of a concerted effort to
correctly attribute ideas; insufficient discussion of related work was considered the major flaw of a previous submission.
With the additional 9th page, we shall restore helpful explanations and full equations for LDSTACK. Similarly, we
agree the experiments have condensed presentation, and deserve more details on the 9th page.

**Discretization error**. In order to prove correctness of LDSTACK, it is not necessary to pass to the original continuous-
time scheme of Banks et al. In discrete time, the convergence of $s_t^{(i)}$ (uniformly across $t$) occurs at $i \leq T$. Once $s_t^{(i-1)}$
is correct (i.e. matches the nonlinear RNN), then $p_t^{(i-1)}$ perfectly corrects $s_{t+1}^{(i)}$. The first layer gets the first state correct,
which makes the second layer get both the first and second states correct, and so on. (This is the "simple recursion" we
briefly mentioned, and will of course elaborate with the available space.)

**Long-term memory**. Our MNIST experiment has short/medium-term dependencies. For long-term dependencies, it is
useful to constrain $A$ to have unit eigenvalues (as in an orthogonal or unitary matrix.) This constraint is trivial within
our framework. Suppose the LDS eigenvalue $\lambda$ has polar representation $(r, \theta)$. Then a real zero part of $\ln \lambda = \ln r + \theta i$
corresponds to eigenvalue magnitude $r = 1$. So, optimize over the $\theta$ of $\ln \lambda$ (with zero real part) rather than the real and
imaginary parts of $\lambda$.

As shown in Figure 2 (top right), this solves the copying memory problem (Arjovsky et al., 2016). The goal is to
remember the first 10 entries $r$ of the input sequence, withhold output for $T$ steps (for which the inputs are just "blanks"),
and, upon seeing a "go" input at time $T + 10$, to output $r$. Unitary RNNs solve the problem, whereas standard RNNs do
not outperform a trivial baseline. There is an LDS which achieves zero error (Henaff et al., 2016), so we don't consider
multiple-layer LDSTACK. Arjovsky et al. use LSTM, simple tanh RNN, and uRNN of respective sizes $n = 40, 80$, and
128 for parameter counts of roughly 6500. We use $n = 160$ for the LDS, which has just 3380 parameters. The $\theta$ are
initialized uniformly at random within $[-2\pi, 2\pi]$. We use the AdaMax optimizer with step size 0.01 and batch size 256
for 300 epochs on a sample size of 10,000. Our solution is the **state of the art**: it uses the simplest (linear) RNN with
the fewest parameters to solve $T = 2000$, which demands full-capacity uRNNs (Wisdom et al., 2016) or later models.

**Numerical stability**. The LDS are indeed unstable at some $\lambda$. However, our initialization suggestion ($\lambda$ as the roots
of a monic polynomial with random coefficients) empirically avoids instability, even without techniques like gradient
clipping. The calculation $\log B_i' = -\sum_{j \neq i} \log(1 - \lambda_j/\lambda_i)$ avoids high-degree powers and is (empirically) stable when
all $\lambda_i \neq \lambda_j$ for $i \neq j$. This empirical success merits more formal/theoretical investigation.

**Line 461 in Supplemental**. Controllability indices are not affected by any *full rank* $U$ transformation, and neither is
normality of the induced distribution. Thank you for pointing this out, along with the other fixes.

[Meta-Review · NeurIPS 2020]

The paper proposes to use linear dynamical systems as a core primitive for building recurrent networks that are more interpretable and easier to analyze. The paper received mixed reviews (below acceptance, top 15% -> top 50%, and top 50%). On the positive side, the approach is novel and the theory well developed. On the negative side, some parts are not clear and the experiments need improvement. Overall, the paper is a stepping stone toward developing recurrent architectures that are better connected with classical dynamical systems, more interpretable and easier to analyze. Thus, I believe this paper offers an interesting direction for future research in machine learning.